# Claudin-2: Roles beyond Permeability Functions

**DOI:** 10.3390/ijms20225655

**Published:** 2019-11-12

**Authors:** Shruthi Venugopal, Shaista Anwer, Katalin Szászi

**Affiliations:** Keenan Research Centre for Biomedical Science of the St. Michael’s Hospital and Department of Surgery, University of Toronto, Toronto, ON M5B 1W8, Canada

**Keywords:** claudin-2, tight junctions, epithelium, cancer, inflammation, fibrosis, paracellular permeability, proliferation, migration

## Abstract

Claudin-2 is expressed in the tight junctions of leaky epithelia, where it forms cation-selective and water permeable paracellular channels. Its abundance is under fine control by a complex signaling network that affects both its synthesis and turnover in response to various environmental inputs. Claudin-2 expression is dysregulated in many pathologies including cancer, inflammation, and fibrosis. Claudin-2 has a key role in energy-efficient ion and water transport in the proximal tubules of the kidneys and in the gut. Importantly, strong evidence now also supports a role for this protein as a modulator of vital cellular events relevant to diseases. Signaling pathways that are overactivated in diseases can alter claudin-2 expression, and a good correlation exists between disease stage and claudin-2 abundance. Further, loss- and gain-of-function studies showed that primary changes in claudin-2 expression impact vital cellular processes such as proliferation, migration, and cell fate determination. These effects appear to be mediated by alterations in key signaling pathways. The specific mechanisms linking claudin-2 to these changes remain poorly understood, but adapters binding to the intracellular portion of claudin-2 may play a key role. Thus, dysregulation of claudin-2 may contribute to the generation, maintenance, and/or progression of diseases through both permeability-dependent and -independent mechanisms. The aim of this review is to provide an overview of the properties, regulation, and functions of claudin-2, with a special emphasis on its signal-modulating effects and possible role in diseases.

## 1. Introduction

### Claudin-2 and the Claudin Family of Tight Junction Proteins

Epithelial cells maintain tissue homeostasis by forming a protective layer to separate the internal milieu from the outside. The epithelium mediates highly regulated directional transport and thus controls the exchange of water and solutes. Tight junctions (TJ) are found at the apical side of the junctional complexes that connect epithelial cells. Their primary role is to generate cell polarity (referred to as fence function), control paracellular transport (gate function), and provide signaling input for a wide variety of cellular events (reviewed in [1,2,3]). The TJs are comprised of a multitude of transmembrane and intracellular proteins. The claudin family of TJ membrane proteins consists of 26 (human) or 27 (rodents) small (20-25 kDa) tetraspan proteins expressed in a tissue-specific manner in epithelial and endothelial cells [4,5,6]. The number of claudins varies between species, and in some species e.g., fish there are a much higher number of claudins, probably due to a larger need for osmoregulation. In this review, we only focus on the mammalian claudin-2. Tsukita and colleagues found that the mouse and human orthologs mostly clustered together, except for claudin-13, which is absent in humans [6,7,8]. 

Claudins form the backbone of the TJ strand and determine junctional permeability. Based on permeability properties, members of the claudin family can be classified into channel forming and sealing proteins [2,4,9,10]. The composition of the TJ strand, i.e., the types and ratio of claudins expressed, determines the permeability of the TJs, which can be altered by selectively replacing specific claudins [11]. 

Claudin-2 was one of the first claudins identified by Tsukita and co-workers as a member of a new class of TJ transmembrane proteins that share tetraspan topology with occludin but have no homology with it [12,13]. Subsequent studies revealed that claudin-2 has a high degree of homology with claudins 1–10, 14, 15, 17 and 19, a group referred to as classic claudins [10]. The mouse and human orthologs of claudin-2 share 70% sequence homology, while their promoters possess a homology of 84% [14,15]. Claudin-2 is a cation-selective channel forming TJ protein expressed in leaky epithelia [16,17,18,19]. In this review, we will refer to paracellular channel formation by claudin-2 as its *permeability function*. Importantly, expression of claudin-2 is dynamically modulated by a variety of conditions, and compelling evidence now indicates that altered claudin-2 expression affects vital biological processes, such as proliferation, migration and cell fate decision. These effects cannot be explained by the permeability function of claudin-2, and they appear to be mediated by specific signaling pathways and transcription factors. This newly emerging, incompletely understood role of claudin-2 will be referred to as its signal-modulating function and will be the focus of this review. In fact, such a novel role is in line with the emerging non-classical functions described for several other claudins [20,21,22].

We will first provide an overview of the properties of claudin-2, and the key inputs regulating its expression. This will be followed by a discussion of its functions including the mounting evidence of its signal-modulating function, and its link to various diseases. 

## 2. Properties of Claudin-2

### 2.1. Expression

Under physiological conditions, claudin-2 mRNA is enriched in the kidneys, where it is exclusively expressed in the proximal tubules, and in the gastrointestinal tract, where the highest expression was reported in the small intestine, liver, gall bladder, and pancreas [23,24,25]. It is also detectable at lower levels in the stomach and colon [24]. Interestingly, claudin-2 expression was also reported outside of epithelia, e.g., in endothelial cells [26,27] and macrophages (e.g., [28]), but it is not clear whether this represents a physiological localization. As discussed later (see Section 5), pathological conditions affect claudin-2 levels, and may also alter its expression pattern, therefore these observations might indicate pathological expression. Indeed, interleukins and Transforming Growth Factor β1 (TGFβ1) were shown to induce claudin-2 expression in macrophages [28]. Whether claudin-2 has a role in non-epithelial cells remains undetermined.

### 2.2. Structure and Interactions

Claudin-2 is a 230 amino acid transmembrane protein, with a calculated molecular mass of 24.5 kDa. Alternatively spliced transcript variants with different 5’ untranslated regions have been found for the claudin-2 gene, although their relevance remains unknown. Like other members of the family, claudin-2 is a tetraspan membrane protein consisting of two extracellular domains (ECL1 and ECL2), a small intracellular loop connecting the second and third transmembrane sections and short intracellular N and C terminal portions (Figure 1) ([12,13] and see [5,10] for review on claudin structure). ECL1 (amino acids 29–81), is responsible for paracellular charge selectivity and permeability [29]. The narrow fluid-filled pores of the claudin-2 channel are lined by polar side chains of ECL1 [16,29,30,31,32]. Mutagenesis studies revealed that the cation specificity of the channel is due to electrostatic interactions between the transported cations and the negatively charged carboxyl sidechain of D65 and the aromatic residue of Y67 [16,31]. These residues allow cations to permeate in a fully or partially dehydrated state. Intramolecular cis interactions between C54 and C64, two highly conserved cysteines in ECL1 stabilize the TJ pore [33]. 

Claudin-2 forms both homo- and heterotypic adhesions, via cis (i.e., molecules in the same cell) and trans (between molecules in neighboring cells) interactions. Cis homodimerization of claudin-2 requires the transmembrane domains and may have a role in organizing the TJ strands [36]. On the other hand, trans homophilic interactions are thought to promote cell-cell adhesion (e.g., [37]). Atomic force microscopy studies probing properties of the interactions between the extracellular loops of claudin-2 revealed that ECL1 was sufficient for homophilic trans interaction, and ECL2 did not mediate these [38]. This is different from what was shown for some other claudins (e.g., claudin-5) [39], where ECL2 had a role in trans interactions, and the reason for such difference remains unknown. Thus, the exact role of the claudin-2 ECL2 is not yet established, since it was not found to be involved in determining permeability or homotypic interactions. Of note, however, a peptide (DFYSP) mimicking a highly conserved region in claudin-2, induced endocytosis, and degradation of the protein [40]. This finding suggests that ECL2 might be key for maintaining claudin-2 at the membrane through a yet to be revealed mechanism.

Some heterotypic trans interactions of claudin-2 have also been described. Claudin-2 can bind to claudin-3 but not to claudin-1 on neighboring cells [41]. However, to date, there is limited information on these interactions, and the exact structural basis for the trans interactions of claudin-2 remains to be mapped. Interestingly, an antagonistic relationship was demonstrated between claudin-2 and two other claudins, and this was implicated in cytokine-induced junction reorganization. Fluorescence recovery after photobleaching (FRAP) analysis of GFP-tagged claudin-4 revealed that claudin-2 and -4 compete with one another for residency at the TJs [42]. Further, claudin-8 was shown to displace claudin-2 from the junctions, resulting in elevated transepithelial resistance (TER) [43]. 

The intracellular interactions of claudin-2 are pivotal for its signal-modulating functions. However, we only have a limited understanding of the protein network associated with claudin-2. The last four C-terminal amino acids of claudins correspond to a PDZ domain-binding motif, and in classic claudins, the two C-terminal amino acids (YV) are 100% conserved. Specificity for different PDZ domains is determined by variation in the neighboring amino acids close to the PDZ domain-binding motifs [44]. In claudin-2, the last 4 amino acids are T-G-Y-V (N- to C-term). This PDZ-binding motif was shown to associate with a variety of TJ plaque scaffold proteins including the membrane-associated guanylate kinase (MAGUK) family adapters ZO-1-3 [45] (Figure 1). Recently, Tabaries et al. identified several new candidate claudin-2 binding partners [34]. These will be described in more detail in Section 4.4. 

The intracellular portions of claudin-2 also undergoes post-translational modifications including phosphorylation [46,47,48], sumoylation [49], and nitration [50] (see Section 3.4). The effects of posttranslational modifications are not fully understood. 

## 3. Regulation of Claudin-2

### 3.1. Context-Dependent Regulation of Claudin-2

The TJs are continuously remodeled in response to environmental cues. In line with this, claudin-2 expression is dynamically modulated by a multitude of inputs in a context-dependent manner that affect both synthesis and turnover (Figure 2). Protein half-life and trafficking are fine-tuned by a variety of post-translational modifications and interactions with a large array of proteins. Many of the pathways that control claudin-2 are dysregulated in various diseases, resulting in disease-associated changes in claudin-2 expression (see Section 5). In general, studies to decipher the mechanisms controlling claudin-2 were performed in cell lines, and it is important to emphasize that significant context-dependent discrepancies are abundant in the literature. Specifically, some treatments were found to be stimulatory in one cell type, but inhibitory in another. Some of these differences may reflect the complex and likely organ-specific regulation of the protein. In support of this notion, the claudin-2 promoter was affected in the opposite direction by EGF in Caco2 colon cancer and in MDCK tubular epithelial cells [51]. However, some caution is warranted when evaluating data in the literature on claudin-2 regulation in cell lines, as the experiments are impacted not only by difference in origin, culture conditions and passage number of the cells, but also a multitude of other parameters. For example, our studies call attention to the importance of cell confluence and junction maturity as a factor in claudin-2 expression regulation. We found that in LLC-PK1 tubular cells, claudin-2 expression was low in subconfluent cultures and increased as confluence was established [52]. Treatment time is another important parameter that is not always taken into consideration when comparing studies. In tubular and intestinal cell lines the effect of TNFα proved to be opposite depending on the length of treatment [53]. Despite these discrepancies in findings, several mechanisms of claudin-2 regulation are now well established.

### 3.2. Signal Transduction Pathways Regulating Claudin-2 Expression

A multitude of signaling pathways, many of which are pro-proliferative and cancer-related, were shown to affect claudin-2 abundance. These pathways affect all aspects of claudin-2 regulation, including the control of synthesis via a set of transcription factors, as well as turn-over and localization of the protein. These latter are affected by post-translational modifications and interactions with various proteins (Figure 2). Many pathways affect multiple aspects of claudin-2 control, and the transcriptional effects are often intertwined with the control of turnover. In this section we will summarize the most studied stimuli and pathways that affect claudin-2 expression, highlighting the mode of their action. The transcription factors affecting claudin-2 will be discussed in more detail in Section 3.3.

One of the best-explored regulators of claudin-2 is the epidermal growth factor receptor (EGFR) and its effectors. Both the Ras/Raf/MEK/ERK and the phosphoinositide 3-kinase (PI3K)/Akt/nuclear factor-κB (NF-κB) pathways were found to target claudin-2 [54,55,56]. EGFR and ERK signaling were shown to affect claudin-2 in many tissues and could mediate effects of a variety of inputs. As mentioned above, both negative and positive effects were reported, highlighting the importance of context (e.g., [51,53,57,58,59,60,61,62]. EGFR can also mediate effects of other stimuli. For example, histone deacetylases including HDAC4 appear to affect claudin-2 via EGFR and ERK signaling [63,64]. Thus, HDAC inhibitors, that are in use or under consideration for therapy may have a major impact on claudin-2. 

The effects of various cytokines on claudin-2 were extensively explored. For example, several interleukins, including IL-1β, 6, 9, 13, and 22 control claudin-2, and some were shown to act via the Jak/Stat pathway [65,66,67]. The effect of TNFα might be organ-specific, as it was verified to have different effects on claudin-2 in intestinal and tubular cells (e.g., [53,56]).

Kinases, downstream from various receptors may alter claudin-2 synthesis, but direct phosphorylation of the protein could also affect turnover (see Section 3.4). For most of the kinases, these details have yet to be established. Src kinases reduced claudin-2 expression [68,69], while protein kinase C (PKC) family members and the stress kinase JNK augmented it (e.g., [70,71,72,73]). Regulation of claudin-2 by Wnt/β-catenin signaling and Rho family small GTPases could also be significant both physiologically and in diseases [52,53,74]. RhoA/Rho kinase and Rac/Pak control both resting claudin-2 levels [52,53,75] and its localization [76]. 

Finally, several miRNAs, including miR-488, miR-16, and the miR-199a-5p/214-3p gene cluster were also implicated in claudin-2 control [54,77,78]. For example, the miR-199a-5p/214-3p gene cluster was found to be downstream from serum response factor (SRF), a pro-fibrotic transcription factor, and was implicated in high glucose-induced claudin-2 downregulation in peritoneal mesothelial cells undergoing epithelial-mesenchymal transition (EMT) [78].

### 3.3. Transcription Factors controlling claudin-2 expression

The TATA-less claudin-2 promoter has binding sites for several transcription factors [14]. Regulation by Caudal-related homeobox (Cdx) homeodomain proteins, hepatocyte nuclear factor (HNF)-1, GATA 2 and -4, AP1, Vitamin D Receptor, TCF/LEF1 and STATs was verified experimentally [14,23,55,72,74,79]. Many of these have significant roles in pathology (see Section 5). HNF-1α and GATA-4 are responsible for claudin-2 expression in specific locations of the gastrointestinal tract, but not in the kidney, highlighting a key organ-specific difference [23,80]. In contrast, GATA-2 was suggested to control claudin-2 in lung and kidney cells [70]. Cdx proteins are intestine-specific transcription factors with significant functions in gut differentiation and carcinogenesis (reviewed in [81]). The claudin-2 promoter is activated by both Cdx1 and 2, which might mediate the effects of the ERK pathway in cancer cells [14,55,82,83,84]. Interestingly, Cdx factors can collaborate with β-catenin-T cell factor (TCF)/lymphoid enhancer factor (LEF) [74] and HNF-1α [14]. The Cdx1 binding site also contains a functional vitamin D response element, which can directly interact with the Vitamin D Receptor transcription factor [79]. This could be significant for intestinal Ca2+ absorption. Finally, the AP1 complex can mediate the effects of JNK and ERK on the claudin-2 promoter [55,60,72,85], while STATs are effectors of interleukins [65,72].

Many stimuli were shown to decrease claudin-2 levels. This downregulation could be due to inhibition of the above signaling pathways and transcription factors. For example, hyperosmolarity was found to reduce claudin-2 promoter activity by inhibiting PKCβ-dependent GATA-2 transcriptional activity [70]. In addition, transcriptional repressors might also act directly on the claudin-2 promoter. Several EMT-inducing transcription factors are known repressors of tight junction protein genes [86,87]. Accordingly, the transcription factor Snail can induce a drop in claudin-2 [88]. Significantly, the exact mechanisms responsible for claudin-2 downregulation remain mostly unclear. For example, we do not yet have a mechanistic understanding of claudin-2 downregulation in tubular cells (Section 5.3), and the reported negative effects of Src kinases remain unexplained [68,69]. Considering the emerging consequences of reduced claudin-2 expression, a better understanding of these mechanisms will be critical.

### 3.4. Claudin-2 Turnover, Trafficking, and Posttranslational Modifications

Claudin-2 is a high turnover protein that dynamically cycles between the membrane and intracellular pools [89]. Claudins are connected to both the microtubules and the actomyosin belt through adapter proteins, and these play key roles in assembly and remodeling of the TJs [90,91,92]. Junctional localization of claudin-2 is not only a prerequisite for permeability functions, but also increases the protein’s half-life (e.g., [52,93]). TJ insertion can be promoted or hindered via interactions with membrane and adapter proteins. As mentioned in Section 2.2. claudin-4 and 8 were shown to exclude claudin-2 from the membrane [42,43]. In contrast, ZO family adapters augment the localization of claudin-2 to the TJs. Silencing of ZO-1 and 2 reduced claudin-2 levels, likely in part by enhancing its degradation [52,67,94,95]. Interestingly, however, loss of ZO-1 also inhibited the claudin-2 promoter, suggesting the existence of feedback regulation between claudin-2 turnover and synthesis [52].

Claudin-2 is retrieved from the TJs by endocytosis. This process was found to be clathrin-dependent in lung and kidney cells [40,48,58]. Other studies have demonstrated a direct binding between claudin-2 and the endocytic scaffold protein caveolin-1 [96,97]. Cytokine-induced gut permeability increase was found to be dependent on caveolin-1 and myosin light chain-dependent endocytosis of TJ proteins [98]. In general, the actomyosin belt plays an essential role in promoting TJ protein retrieval induced by cytokines [91,99], although its specific role in claudin-2 retrieval remains less well established. 

The fate of endocytosed claudin-2 is poorly understood. One study found recycling of the endocytosed protein back to the TJs [89]. Several other studies showed that it is targeted for lysosomal degradation (e.g., [40,47,71]). The small GTPases Rab14 and the atypical PKCι, an essential regulator of apico-basal polarity were implicated in the control of claudin-2 trafficking [71]. The autophagy pathway can also target claudin-2 [100,101]. TNFα-induced inhibition of autophagic degradation was suggested to elevate claudin-2 expression in intestinal cells [102]. TNFα also caused an acute ERK-dependent decrease in claudin-2 degradation in tubular cells [53]. Interestingly, similar to some other claudins, claudin-2 was also shown to translocate to the nucleus [103] (see Section 4.2).

Post-translational modifications appear to be key determinants of claudin-2 localization, and turnover, and likely control protein-protein interactions. The claudin-2 intracellular segment contains several potential phosphorylation sites. Phosphorylation of Y224 in human claudin-2 (referred to in some papers as Tyr-6) was shown to affect the affinity of the ZO-1 PDZ binding domain [35], which may impact trafficking. In many publications, the numbering of the position of the C-terminal claudin residues follows the convention that the C-terminal amino acid is referred to as the P0 residue. Subsequent residues toward the N terminus are termed P−1, P−2, etc. Thus, the tyrosine in question is often referred to as position P-6. The possible importance of this site is supported by the fact that many claudins have a tyrosine at this position [35]. S208 is another site that was shown to be phosphorylated. It was implicated in the control of membrane retention, lysosomal localization, and degradation, as well as nuclear localization [46,48,103]. Accordingly, a non-phosphorylatable mutant (S208A) was found to be more cytosolic and lysosomal, while the phospho-mimetic mutant (S208E) localized more to the plasma membrane. Interestingly, mutants, that prevent claudin-2 from insertion to the TJs were found to be poorly phosphorylated, suggesting that TJ insertion is a necessary step for the protein to become phosphorylated. Since the S208 site was implicated both in membrane retention and nuclear localization [46,48,103], it is conceivable that a coupling between claudin-2 TJ insertion and nuclear localization may exist. Accordingly, S208 phosphorylation (e.g., by PKC) could induce plasma membrane localization of claudin-2, while dephosphorylation of this site might promote nuclear translocation [46,48,103]. 

Claudin-2 also undergoes other post-translational modifications. Sumoylation was found to control membrane localization, ubiquitination, and degradation of claudin-2 [49], and nitration of tyrosine residues was associated with reduced expression [50]. Despite these emerging data, however, we still poorly understand how claudin-2 trafficking and fate are determined. 

Importantly, many details of the complex, multilevel regulation of claudin-2 are closely linked to its pathogenic roles, and therefore an improved understanding of the molecular details will be pivotal for further insight into its dysregulation.

## 4. Functions of Claudin-2

### 4.1. Permeability Functions

Claudin-2 forms paracellular cation and water permeable channels. An elegant study by Weber et al. used a novel trans-TJ patch-clamp technique to show that the claudin-2 channel exhibits symmetrical and reversible conductance of ~90 pS [32]. They found that the channel is gated with one open and two distinct closed states. Claudin-2, on the other hand, is relatively impermeable to uncharged solutes [17,18], but permeable to cations and water [104]. Interestingly, water transport is mediated by the same pore that allows cation transport ([105] and reviewed in [106]). 

These permeability properties are well verified in cell lines and knockout mouse models. Claudin-2 overexpression and silencing in various cell lines decreases and increases TER, respectively [17,18,53,104,107]. Claudin-2 silencing also eliminates preference of the paracellular pathway towards Na^+^ over Cl^−^ [19]. 

Knockout mouse models also revealed that the selective, site-specific localization and cation channel properties of claudin-2 are essential for highly specialized functions in the kidneys and the bile duct (reviewed in [108]). In the kidney, claudin-2 is pivotal for efficient Na^+^ and water reabsorption in the proximal tubules. Accordingly, the S2 segment of the proximal tubules in claudin-2 KO mice showed significantly reduced net transepithelial reabsorption of Na^+^, Cl^−^, and water [109]. Although the mice had normal Na^+^ homeostasis when fed a regular diet, administration of salt revealed Na^+^ handling deficiencies in the kidneys. This suggests that under resting conditions the kidneys could compensate for the loss of the passive claudin-2-mediated Na^+^ transport, but under stress, this capacity was insufficient. This conclusion was also supported by findings from Pei et al. [110]. They showed that in claudin-2 knockout mice, upregulation of active transcellular pathways took over the role of passive paracellular Na^+^ transport. However, claudin-2 appears to be vital for energy efficiency of the proximal tubular Na^+^ reabsorption process, and the higher energy-demand for transcellular transport resulted in medullary hypoxia and increased susceptibility to renal ischemia-reperfusion injury in the claudin-2 KO mice (see also Section 5.3). 

In addition to a significant role in ion transport, studies using KO mice also revealed that claudin-2-mediated water transport contributes 23-30% of total water absorption in the proximal tubules [106]. Taken together, the permeability function of claudin-2 in the kidneys is fundamental for Na^+^ and water homeostasis and blood pressure regulation [106,109,111]. 

In the liver and bile system, claudin-2 is highly expressed in hepatocytes and cholangiocytes and plays a central role in water transport associated with bile generation. Accordingly, in claudin-2 KO mice, the rate of bile flow was found to be reduced by half, resulting in a significant increase in bile concentration, and gallstones [112]. 

In the small intestine, claudin-2 plays a role in luminal Na^+^ homeostasis. Surprisingly, however, it was found to be dispensable for Na^+^-driven nutrient transport [113]. This contrasted with the role of claudin-15, which proved to be indispensable for both luminal Na^+^ transport and Na^+^-driven absorption of vital nutrients. The absence of these two claudins in double knockout mice resulted in severe malnutrition [114]. 

Taken together, claudin-2 plays a key role as a paracellular permeability molecule. However, as discussed in the next sections, emerging evidence suggests that its functions go beyond permeability. 

### 4.2. Role in Proliferation

Evidence is mounting that claudin-2 is not only a paracellular channel protein, but also acts as a signal modulator and integrator (Table 1). This conclusion is derived in part from the correlation between claudin-2 expression and altered outcomes, and in part from overexpression and silencing studies that reveal the effects of altered claudin-2 expression on various outcomes. Importantly, dysregulation of claudin-2 expression appears to be an important event in several diseases (see Section 5). 

As discussed in Section 3.2, many proliferative pathways affect claudin-2 expression, and these are often dysregulated in diseases, including in cancer. In fact, a strong correlation has been established by many studies between proliferation/cell viability and altered claudin-2 expression. For example, the silencing of the TJ adapter protein cingulin in kidney tubular cells simultaneously elevated claudin-2 levels and caused a RhoA-dependent increase in G1/S phase transition [75]. In A549 lung adenocarcinoma cells, ephrinA1 reduced both claudin-2 expression and proliferation, and both were mediated by Cdx-2 [115]. Recently, miR-488 was shown to suppress both claudin-2 expression and cell viability in colorectal carcinoma cells [77]. 

These correlations, however, do not prove a causative connection. More definitive proof is provided by gain- and loss-of-function experiments in a variety of cell lines, showing that primary claudin-2 expression changes can alter proliferation (e.g., [67,103,116]). The experimental evidence connecting claudin-2 with downstream effects is summarized in Table 1. Experimental interventions that reduce claudin-2 expression in lung cancer cells reduce proliferation (see Section 5.1) [64,117]. Thus, claudin-2 might be a permissive factor for proliferation and may act as a mediator of pro-proliferative signaling. Accordingly, silencing claudin-2 in the colon cancer cell line Caco-2 and in tubular cells prevented EGF- and TNFα-induced increase in cell proliferation [51,67]. 

Claudin-2 impacts cell cycle progression and expression of cell cycle regulators both in cultured cells and in a transgenic mouse model [67,103,116]. In a Villin-claudin-2 transgenic mouse model that overexpresses claudin-2 in the colon, the proliferation of colon cells was augmented by claudin-2 overexpression via the PI-3Kinase/Bcl-2 pathway [116]. In lung cancer cells claudin-2 controls the G1/S transition through cyclin D1 and E1 [103]. This effect was mediated by the Y-box transcription factor Zonula occludens 1-associated nucleic acid-binding protein (ZONAB). ZONAB is known to shuttle between the TJs and the nucleus. Interestingly, claudin-2 was also found in the nucleus where it formed a complex with ZO-1 and ZONAB [103]. Moreover, its nuclear levels increased during the G1/S transition. Interaction between claudin-2 and ZONAB was also demonstrated in colon cancer cells, where symplekin, a transcriptional regulator that cooperates with nuclear ZONAB, was shown to control claudin-2 levels [118]. The molecular underpinning of the nuclear translocation of claudin-2 remains insufficiently understood. No classic nuclear localization sequence is recognizable in claudin-2, but mutagenesis studies revealed that the C-terminal cytosolic segment (but not the PDZ domain) was essential for nuclear transport. As discussed in Section 3.4 S208 phosphorylation is a key regulator of nuclear localization, and a phospho-incompetent mutant (S208A) was found to be more nuclear. 

Another cell cycle regulator affected by claudin-2 is p27^kip1^ (cyclin-dependent kinase inhibitor 1B, CDKN1B). The expression of this molecule is high in quiescent cells, where it blocks cell cycle entry. We found that claudin-2 silencing elevated p27^kip1^ levels, and this effect was mediated by the GEF-H1/RhoA pathway [67]. Since p27^kip1^ has tumor suppressor effects, its regulation by claudin-2 might be relevant for carcinogenesis. 

Altered claudin-2 expression affects several other transcription factors too. Sp1, a zinc finger transcription factor belonging to the Sp/KLF family and c-jun, part of the AP1 early response complex were found to be affected by claudin-2. Specifically, claudin-2 knockdown decreased the levels of phosho-c-Jun and reduced nuclear Sp1 [119,120]. Importantly, these transcription factors also regulate cell proliferation (e.g., [121]) suggesting another possible link. 

Recently, claudin-2 was shown to affect miRNAs. This effect was implicated in the control of self-renewal in colon cancer stem cells [122]. Although the mechanisms that connect claudin-2 to miRNA regulation remain undefined, such effects might link claudin-2 to many downstream events. 

Finally, yet another intriguing observation is that claudin-2 can localize along the primary cilium [123]. Claudin-2 is dispensable for ciliogenesis, and its exact role in the cilium remains unclear. However, since the primary cilium is present only in quiescent cells and its disassembly is necessary for cell proliferation [124], this intriguing observation clearly warrants further exploration. 

### 4.3. Migration

The effects of claudin-2 on cell migration may be relevant in both cancer biology and tissue regeneration. Although many studies found that claudin-2 affects migration, significant contradictions are notable among these (Table 1). Specifically, both claudin-2 overexpression and silencing were shown to enhance migration, pointing to context-dependent factors. 

In support of a role for claudin-2 in augmenting motility, in various cancer cells claudin-2 overexpression correlates with enhanced migration. In gastric cancer cells, the Helicobacter pylori virulence factor cytotoxin-associated gene (CagA) augmented both claudin-2 expression and invasiveness [82] (see Section 5.1). In line with a stimulatory role, in several cells, a decrease in claudin-2 abundance reduced migration. Non-steroidal anti-inflammatory drugs (NSAIDs), that may provide protection against cancer, reduced both migration and claudin-2 expression in colon, stomach, and lung cancer cell lines [125]. In these studies, the authors provided further evidence for a causal link, by showing that overexpression of claudin-2 stimulated migration and that claudin-2 re-expression restored migration following drug treatment. 

Underlying mechanisms for a positive effect of claudin-2 on migration are yet to be unraveled, but a few possible players have emerged. Matrix metalloproteases (MMP) might be important mediators of the effect. This family of zinc-dependent proteases contains over 25 members. Several of them were shown to aid migration through the digestion of extracellular substrates [126] and are considered as potential therapeutic targets. Claudin-2 was shown to affect MMP9 activity, although surprisingly claudin-2 decrease and increase exerted a similar augmenting effect [120,127]. Inhibition of A549 cancer cell migration in a wound-healing assay by claudin-2 knockdown was attributed to reduced MMP-9 expression and activity due to Sp1 inhibition [120]. However, this might not be a universal effect, as claudin-2 overexpression-induced augmented migration of Caco-2 cells proved independent of MMPs [128]. 

As opposed to the above-described effects, some studies have found that claudin-2 exerts an inhibitory effect on migration. In MDCK tubular cells inducible knockdown of claudin-2 increased MMP-9 mRNA and activity, and this enhanced migration in a wound-healing assay [120]. Similarly, hyperosmotic stress-induced decrease in claudin-2 expression in the same cell line led to augmented migration, and the phenotype was rescued by re-expression of claudin-2 [70]. Of note, MDCK cells are not derived from cancer, but from normal renal tubular epithelial cells, which might explain the different effects on migration. Claudin-2 loss was also shown to augment migration downstream from the transcription factor Spi-B. This factor is normally restricted to the lymphocyte lineage, but it is frequently expressed in lung cancer. Further, it was shown to repress the transcription of claudin-2, which led to enhanced invasiveness [129]. Another study suggested that in osteosarcoma cells claudin-2 loss augmented migration via the afadin/ERK pathway [130]. Along the same lines miR-488 was shown to regulate invasion and lymph node metastasis in colon cancer cells through claudin-2-dependent control of the MAPK pathway [77]. 

Another mechanism whereby claudin-2 might affect migration is through effects on EMT [131,132]. This process involves a coordinated genetic reprogramming, during which cells lose their epithelial characteristics and gain mesenchymal properties, including enhanced motility. This process is key during embryonic development and wound healing, but also plays a role in cancer metastasis formation. Tight junction downregulation is a hallmark of full-blown EMT [86], but interestingly, multiple studies now suggest a reciprocal correlation too, namely active EMT-regulating roles for claudins. For example, claudin-1 overexpression was shown to promote EMT, while claudin-3 suppressed it [133,134]. While the exact role of claudin-2 in this process remains to be fully established, we recently found that claudin-2 silencing enhanced RhoA-mediated activation of Myocardin-Related Transcription Factor (MRTF), and upregulated Slug, two pro-EMT transcription factors [67]. The potential role of these in altered migration following claudin-2 loss remains unknown. 

Taken together, claudin-2 was shown to have both a positive and a negative effect on migration, suggesting a need for optimal claudin-2 levels for efficient migration regulation. Clearly, further mechanistic inquiries will be required to establish the context-dependent mechanisms and importance of these effects.

### 4.4. Signal-Modulating Effects of Claudin-2: Emerging Mechanisms Underlying Roles in Biological Processes 

As discussed in Section 4.2 and Section 4.3, evidence is mounting that claudin-2 affects cell behavior (Table 1). However, a coherent picture of the underlying mechanisms has yet to emerge. Many studies suggest that claudin-2 may be a signal modulator and integrator protein. Nevertheless, an in-depth understanding of how claudin-2 is linked with altered signaling pathways remains elusive. 

The TJ cytoplasmic plaque contains a large array of signaling and regulatory proteins. These are connected to the TJ membrane proteins by various adapters. For many of these proteins, recruitment to the junctions results in inactivation, or a spatial restriction of activity (for review see [90,136,137]). The specific role of individual claudins in the organizing of the cytoplasmic plaque is not yet resolved. As mentioned in Section 2.2, the cytoplasmic tail of claudin-2 interacts with multiple proteins, but the claudin-2 interactome has not yet been fully mapped, and context-dependent regulation of the interactions remains unknown. As mentioned earlier, a recent study from the Siegel group identified some candidate binding partners of the claudin-2 PDZ domain [34]. Using metastatic breast cancer cells with elevated claudin-2 expression, they demonstrated that the last three C-terminal amino acids of claudin-2 (comprising the PDZ-binding motif) are necessary for anchorage-independent growth. They identified several potential partners of this motif, including afadin, which is an actin filament- and Rap1 small GTPase-binding protein encoded by the MLLT4/AF-6 gene. Loss of afadin impaired colony formation of breast cancer cells in soft agar and reduced lung and liver metastasis, verifying the significance of the claudin-2/afadin complex. Importantly, some of their findings suggested the existence of afadin-independent mechanisms too. The identified potential other partners, including the polarity protein Scrib, the small GTPase regulator Arhgap21, PDZ-LIM domain-containing proteins (PDLIM) 2 and 7, and the exocytosis-controlling Rims-2 (regulating synaptic membrane exocytosis protein 2) may mediate such downstream effects. However, whether these are direct binding partners or interact via adapters, and their potential functional significance remains to be established. 

The crosstalk between claudin-2 and the cytoskeleton may play a central role in mediating downstream effects. Claudin-2, as all TJ membrane proteins, is anchored to the junctional actomyosin belt and microtubules through adaptor proteins. Among these, ZO family proteins directly bind claudins, and connect to F-actin either directly, or indirectly through actin-binding proteins [138,139]. Thus, it is conceivable that altered claudin-2 expression might affect the organization of the prejunctional cytoskeleton. In support of this, we recently found that in tubular cells, claudin-2 silencing altered acto-myosin organization through RhoA [67]. 

The microtubules also play a role in junction regulation and might be affected by the TJ organization. Cingulin and its paralog paracingulin interact with TJ proteins, likely through ZO adapters, and thus connect to actin. Cingulin also binds the microtubules and has a role in organizing the microtubular planar apical network in mammary epithelial cells [140,141]. Further, association of cingulin with actin filaments and the microtubules was found to be regulated by phosphorylation and plays a crucial role in the control of the epithelial barrier [141]. The specific roles of individual claudins as organizers of the cytoplasmic plaque complexes and their impact on the cytoskeleton are yet to be uncovered. Nevertheless, it is conceivable that some of the signal modulating effects of claudin-2 is mediated by an impact on the acto-myosin or the MTs. 

Interestingly, claudin-2 appears to have a bidirectional relationship with the cytoskeleton and some signaling pathways. Specifically, some of the pathways modulated by claudin-2 can also regulate its expression. This suggests the existence of various regulatory feedback cycles (Figure 2). Such regulatory loops also pose a challenge for defining clear “upstream” and “downstream” events. The ERK and Rho signaling pathways are prominent examples of this complexity. The role of MEK/ERK signaling in basal and stimulus-induced claudin-2 expression is well documented [53,55,60,61,85,142,143]. Besides, a recent study reported the converse effect: claudin-2 silencing was found to augment MEK/ERK1/2 signaling [130], an effect that may help to restore claudin-2 levels. A complex relationship also exists between Rho family small GTPases and claudin-2. We found that RhoA/Rho-kinase are negative regulators of claudin-2 expression in cultured tubular cells [53]. On the other hand, as mentioned above, in the same cells claudin-2 silencing enhanced RhoA activity through the exchange factor GEF-H1, suggesting that in resting cells, claudin-2 might suppress RhoA activation [67]. Thus, claudin-2 loss can augment Rho activity, which in turn may further reduce claudin-2 expression, in a possible self-augmenting cycle. 

## 5. Claudin-2 in Diseases

A growing number of studies document altered claudin-2 expression, phosphorylation and/or localization in various pathological conditions. Initially, the described claudin expression alterations were considered an epiphenomenon, i.e., they were assumed to arise due to the underlying disease process but were thought not to be causally contributing to pathogenesis. Nevertheless, interest in pathological alterations in claudin-2 expression was boosted by the hope that it can be used as a diagnostic and/or prediction marker. As described in Section 4, in the past years, strong evidence accumulated in support of a causal link between claudin-2 dysregulation and functional alterations, the key points of which are the following. First, signaling pathways that are known to be overactivated in diseases can alter claudin-2 expression, and a good correlation exists between disease stage and claudin-2 expression. Second, loss-of-function and gain-of-function studies recapitulate some aspects of the functional changes. Thus, primary changes in claudin-2 expression can alter cell behavior. The mounting experimental evidence prompted a paradigm shift, favoring a pathogenic role for claudin-2. Although dysregulation of claudin-2 is likely not a primary cause, pathological changes in claudin-2 abundance and/or localization might play significant roles in the generation, maintenance and/or progression of diseases.

In the following sections, we will provide an overview of the evidence linking claudin-2 to cancer, and various non-malignant pathologies, such as inflammatory gastrointestinal and kidney diseases.

### 5.1. Claudin-2 in Cancer and Metastasis Formation

An expanding body of literature documents dysregulated claudin-2 expression in gastric, colorectal, lung, breast, and renal carcinomas and in osteosarcoma (e.g., [51,144,145,146,147,148]. As described in Section 4.2 and Section 4.3, claudin-2 levels affect processes underlying carcinogenesis and metastasis formation, including proliferation, migration and epithelial-mesenchymal transition. The following overview however also highlights that the role of claudin-2 is likely complex, and differences in its impact might exist not only based on tissue origin but also cancer stage. 

*Cancers of the gastrointestinal tract*—Claudin-2 is highly expressed in gastric and colorectal cancers and its expression level shows a good correlation with the development of these tumors. For example, a gradual increase was observed in claudin-2-positivity in various stages of gastric carcinogenesis from no expression in gastritis to elevated expression in dysplasia and gastric intestinal-type adenocarcinoma [149]. In fact, in this study, 73% of adenocarcinoma samples were found to be claudin-2 positive.

As described in Section 3.2 and Section 3.3, claudin-2 is the target of key signaling pathways and transcription factors central to gastrointestinal cancers. One prominent example is the significant correlation between Cdx2 and claudin-2 expression in gastric dysplasia and cancer [150]. Further, a potentially important link between claudin-2 and H. pylori was also uncovered. H. pylori infection shows a strong correlation with gastric cancer [151], and the virulence factor CagA is considered a gastric oncogene. During infection, this protein is translocated into and reprograms the gastric cells. Interestingly, CagA augmented claudin-2 expression in gastric cells, and promoted invasiveness through effects on Cdx2 [82]. 

Hyperactivation of Wnt signaling and the consequent increase in gene transcription by TCF/ LEF is a hallmark of colon cancer (reviewed in [152]). Wnt-1 was shown to increase claudin-2 promoter activity through β-catenin/LEF-1 [74]. Increased claudin-2 and β-catenin expression were detected in active inflammatory bowel disease (IBD), adenomas, and IBD-associated dysplasia, but not in acute, self-limited colitis [153]. These data suggest that β-catenin might mediate an increase in claudin-2 during carcinogenesis. 

EGFR signaling is yet another key pathway linking colon cancer and claudin-2. EGFR activation upregulated claudin-2 in colon cancer cells, and claudin-2 expression was decreased in the colon of waved-2 mice that have EGFR activation deficiency [51] (see also Section 3.2 on the effects of EGFR on claudin-2). 

Taken together, these studies suggest that the initial upregulation of claudin-2 could be due to the overactivation of the above-discussed pathways, and this may promote carcinogenesis. To prove a causal link between elevated claudin-2 and the properties of cancer, exogenous claudin-2 was overexpressed in colorectal cancer cell lines. An increase in claudin-2 expression promoted colonocyte proliferation and anchorage-independent colony formation and stimulated tumor formation in colorectal cancer xenografts (e.g., [51,116]). Further, claudin-2 expression was a negative predictor for post-chemotherapy disease-free survival of colon cancer patients [51]. 

Another key role of claudin-2 could be its effect on colorectal cancer stem-like cells [122]. Claudin-2 promoted self-renewal of these cells via miRNAs and the hippo effector yes-associated protein (YAP). Interestingly, claudin-2 was also detected in human colorectal cancer-associated fibroblasts (CAFs). This finding is especially intriguing since claudin-2 is regarded as an epithelial molecule. Its expression in CAFs was linked to KRAS mutation status and correlated with reduced progression-free survival [154]. Of note, claudin-2 was also shown to be enriched in macrophages associated with mammary tumors [28]. Thus, an attractive hypothesis is that claudin-2 may increase proliferation/survival and migration of fibroblasts and macrophages in the tumor environment.

*Lung cancer*—Primary lung cancer is among the deadliest types of tumors worldwide, and adenocarcinomas are the most frequent forms of non-small cell lung cancer. Claudin-2 expression in normal bronchial epithelium is low [85]. In contrast, according to one study, two-thirds of examined lung adenocarcinoma samples overexpressed claudin-2 [145]. Surprisingly, in these cells, claudin-2 was found mostly in cytoplasmic granules. Adenocarcinoma is often associated with upregulated EGFR activity and KRAS/MEK/ERK signaling. Indeed, EGFR signaling is key in upregulating claudin-2 in the human lung adenocarcinoma cell line A549 [60]. These cells were used in a series of studies to demonstrate that claudin-2 may be a therapeutic target in lung cancer. The studies showed that rapid proliferation of A549 cells required claudin-2. Accordingly, claudin-2 knockdown impaired cell growth and migration [103,120] and elevated sensitivity for anti-cancer agents [119]. Importantly, a variety of interventions, including epigenetic inhibitors and various flavonoids reduced claudin-2 expression and decreased proliferation. These effects were counteracted by retransfection of claudin-2 [64,117]. Claudin-2 abundance can also be reduced by a peptide that mimics a sequence in ECL2. This peptide-induced endocytosis and lysosomal accumulation of claudin-2, resulting in lysosomal damage and necrotic cell death [40]. Taken together, these studies provided strong evidence that targeting claudin-2 can mitigate proliferation and raised hope that this approach will prove useful in cancer therapy (see further elaboration in Section 5.4). 

*Osteosarcoma and breast cancer*—Interestingly, in some tumors, claudin-2 expression is decreased. Osteosarcoma cells have lower claudin-2 expression than normal osteoblasts [130], and this is associated with enhanced migration. In breast cancer, increased or decreased claudin-2 expression might be associated with different cancer types and stages [155,156]. As discussed below, in this type of tumor, claudin-2 expression appears to correlate with metastasis formation. 

*Metastasis*—The importance of claudin-2 in metastasis is also emerging. Claudin-2 might affect migration via a variety of effects. Its possible role in migration was discussed in Section 4.3. Beyond general effects on migration, claudin-2 also appears to affect metastasis through mediating specific adhesion of the metastatic cells, thereby promoting invasion into distant organs. Interestingly, the exact role of claudin-2 in metastasis appears to vary based on the site of invasion. In some breast cancer reduction in claudin-2 was associated with lymph node metastasis and higher clinical stage [156]. In contrast, claudin-2 overexpression was shown to augment breast carcinoma metastasis to the liver. Claudin-2 expression is elevated in liver metastases compared to primary breast cancer cells [157]. Indeed, claudin-2 promoted attachment and survival of metastatic cells specifically in the liver and enhanced their interactions with hepatocytes [37]. Due to these effects, claudin-2 was found to be a negative prognostic factor that predicts liver metastasis in breast cancer [158]. Metastatic breast cancer cells showed a general increase in adhesion to the extracellular matrix, as claudin-2 promoted surface expression of α2β1- and α5β1-integrins [157]. However, while these effects might contribute to invasiveness in general, enhanced interaction between hepatocytes and breast cancer cells proved to be independent of integrins. Instead, the effect required the first claudin-2 extracellular loop (ECL1) and was likely attributable to homotypic trans interactions between claudin-2 molecules. Thus, claudin-2 in the metastatic breast cancer cells can bind to claudin-2 on hepatocytes, thereby promoting their adhesion to the liver [157]. Further, the PDZ-binding motif (YV) in claudin-2 was found to be necessary for anchorage-dependent growth and metastases [34]. These exciting findings raise hope that blocking peptides or neutralizing antibodies against the identified domain might be effective in reducing liver metastasis (see Section 5.4). 

### 5.2. Claudin-2 in Gut Inflammation

Claudin-2 emerged as an important factor in IBD, a term describing conditions such as ulcerative colitis and Crohn’s disease. These chronic diseases are hallmarked by compromised epithelial barrier and are thought to be caused by a dysregulated immune response in the gut (e.g., reviewed in [159,160]). Increased paracellular permeability is likely a critical pathogenic factor. Specifically, the disease appears to be maintained by a vicious cycle: increased epithelial permeability due to dysregulation of TJ proteins elicits an aberrant immune response, and the ensuing inflammation further disrupts TJs and aggravates the condition. The effects of inflammation on epithelial permeability have been long known. Elevated claudin-2 expression in IBD is attributed to the presence of cytokines (e.g., [56,161,162] and reviewed in [159,163]). The permeability function of claudin-2 appears to be central in IBD, although a causal link is hard to decipher. Studies in transgenic mice explored and verified the role of the permeability function of claudin-2. In contrast, little is known about the possible significance of claudin-2 as a signal modulator in IBD. Somewhat surprisingly, recent studies point to a possible protective role of claudin-2 against injury. A transgenic mouse with targeted overexpression of claudin-2 in the colon was protected against colitis-associated injury [116]. This study also highlighted the complex roles of claudin-2 in intestinal homeostasis and IBD. Claudin-2 overexpression in the colon augmented mucosal permeability but did not by itself cause inflammation. Instead, the mice had longer colons and elongated crypts, a result of accelerated colonocyte proliferation. Most notably, despite the leaky colon, the mice were significantly protected against experimental colitis, and inflammation-associated gene expression was sharply downregulated. Some of the effects could be the result of reduced apoptosis and augmented regeneration due to faster proliferation. Thus, in this respect, claudin-2 elevation may at least initially confer some protection in IBD. This notion is also supported by another study that used claudin-2 null mice to explore the effects of intraperitoneally injected TNFα and experimental colitis [135]. The data revealed augmented colorectal inflammation in the KO mice compared to WT animals. These effects were mediated by NF-κB signaling. The mice also had increased expression of IL-6 and IL-1β and higher intestinal myosin light chain kinase levels. Overall, these studies suggest that increased claudin-2 expression might suppress inflammatory signals. Importantly, although the proliferative effect observed could be beneficial for regeneration, this is a double-edged sword. IBD elevates the risk for developing colorectal cancer and augmented claudin-2 expression could contribute to this. As mentioned in Section 5.1, claudin-2 levels rise with the development of dysplasia and cancer. This raises the intriguing possibility that elevated claudin-2 expression might be a crucial connection between inflammation and cancer. Thus, while elevated claudin-2 might protect against injury and enhance regeneration, it could raise the risk of cancer. 

The role of claudin-2 in intestinal diseases might go beyond IBD, as it was also implicated in diseases caused by enteropathogenic bacteria. Salmonella invasion elevated claudin-2 protein and mRNA levels both in cultured cells and in the intestine of mice. Upregulation of claudin-2, in turn, elevated permeability and promoted internalization of the bacteria [73]. Finally, recent studies raise the possibility that claudin-2 might contribute to chronic pancreatitis, a progressive inflammatory disease that is frequently associated with alcohol consumption. A genome-wide association search found that polymorphisms of the claudin-2 locus confer an increased risk of alcohol-induced pancreatitis [164]. Since claudin-2 is essential for bile formation, it will be interesting to see if claudin-2 polymorphism might also predispose for gall bladder disease and gallstones.

### 5.3. Claudin-2 in Kidney Disease

Claudin-2 mRNA levels are highest in the kidney, where it is a key mediator of cation and water transport in the proximal tubules (reviewed in [165,166,167]). Although a direct role of claudin-2 in kidney disease has not been definitively established, several observations suggest a possible link. 

Claudin-2 is affected by inflammatory cytokines in tubular cells. Specifically, short-term TNFα treatment caused an increase in claudin-2 abundance [53]. In contrast, prolonged (>8h) TNF-α or IL-1β treatment downregulated claudin-2 expression [53,67]. Interestingly, a large variety of potentially harmful stimuli can downregulate claudin-2 expression in tubular cells. In addition to these cytokines, metabolic acidosis [168], changes in osmolarity [48,127], oxidants [169], EGF [58,170], sheer stress [171], the flavonoid quercetin [172], and drugs, such as the immunosuppressants sirolimus and cyclosporine A [173], reduce claudin-2 levels. Reduced claudin-2 expression was also recorded in various kidney disease animal models, including cisplatin-induced nephrotoxicity [174], diabetic nephropathy [50] and obstructive nephropathy-induced fibrosis [67]. These findings are of significance because claudin-2 was found to exert a protective effect against kidney injury by reducing the energy demand of transport processes [110]. Further, as mentioned above loss of claudin-2 in tubular cells induced RhoA activation, and claudin-2 knockout mice had elevated RhoA activity [67]. This led to reduced proliferation and activation of MRTF, a transcription factor that mediates profibrotic epithelial transition [67,175]. Thus, reduced claudin-2 expression in stress conditions might increase susceptibility for injury and may promote fibrosis. 

Finally, the proximal tubular paracellular pathway also plays a central role in Ca2+ reabsorption. Accordingly, claudin-2 null mice were hypercalciuric, a condition that enhances kidney stone formation [109]. Thus, conditions that reduce the abundance of claudin-2 in the kidney might also lead to kidney stone disease, a condition affecting a significant portion of the population.

### 5.4. Development of Therapies Targeting Claudin-2

Several studies raise hope that targeting claudin-2 could have beneficial effects. Indeed, in a proof of principle study, a monoclonal antibody that recognizes the first extracellular loop (ECL1) of claudin-2 was shown to prevent TNFα-induced TJ disruption [176]. The same group recently generated a human-rat chimeric monoclonal antibody against claudin-2 and showed that it accumulated in claudin-2 positive sarcoma xenografts. The antibody also suppressed tumor growth [177]. Importantly, no major adverse effects were found. 

Taken together, accumulating evidence supports that claudin-2 can be considered a diagnostic and prognostic marker in various diseases and is an exciting potential therapeutic target.

## 6. Open Questions

Claudin-2 is now established as a pathogenic factor in several diseases due to its permeability-dependent and -independent functions. The recognition that claudins have major signal modulator, permeability-independent functions reenergized claudin research, leading to an explosion in the number of studies published. Although the fact that altered claudin-2 expression affects signaling and cell behavior is firmly established, many questions remain unanswered. It is worthwhile to articulate some of these as they set the direction for future research. Importantly, our understanding of mechanisms mediating pathological dysregulation of claudin-2 is still incomplete, which precludes the design of interventions. There are also fundamental unanswered questions regarding the mechanisms that link claudin-2 to various downstream events. We do not know the role of claudin-2 localization in its signal-modulating effects. Are these effects mediated by TJ localized claudin-2? Given that claudin-2 was shown to reside in other compartments (cytosolic vesicles, nucleus), some of the non-classical effects may well be mediated by these claudin-2 sub-pools. Further, many studies revealed the importance of altered claudin-2 expression in the downstream effects, but we do not know whether these are exclusively related to the number of molecules or if further regulation (e.g., by distinct posttranslational modifications) are also involved. Extra- and intracellular binding partners of claudin-2 are likely crucial mediators of signal-modulating effects. Indeed, TJ-localized claudin-2 can interact with other membrane proteins and may act as a sensor of neighboring cells. However, it is not clear whether claudin-2 needs to be engaged for its signal-modulating effects. Further, the claudin-2 interactome is only now starting to emerge and uncovering the context-dependent regulation of interactions will be key for a better understanding. Adapters in the TJ plaque generate signaling complexes, that may change depending on the availability of claudin-2, leading to recruitment or release of signaling intermediates and alterations in the cytoskeleton organization. However, the specific role of claudin-2 in such events remains unknown. Ongoing research from many groups will likely help fill these knowledge gaps leading to a better mechanistic understanding of the signal-modulating effects of claudin-2. The emerging knowledge has already informed the design of interventions targeting claudin-2. Such future therapies have the potential to benefit many patients with a broad spectrum of diseases. 

## Figures and Tables

**Figure 1 ijms-20-05655-f001:**
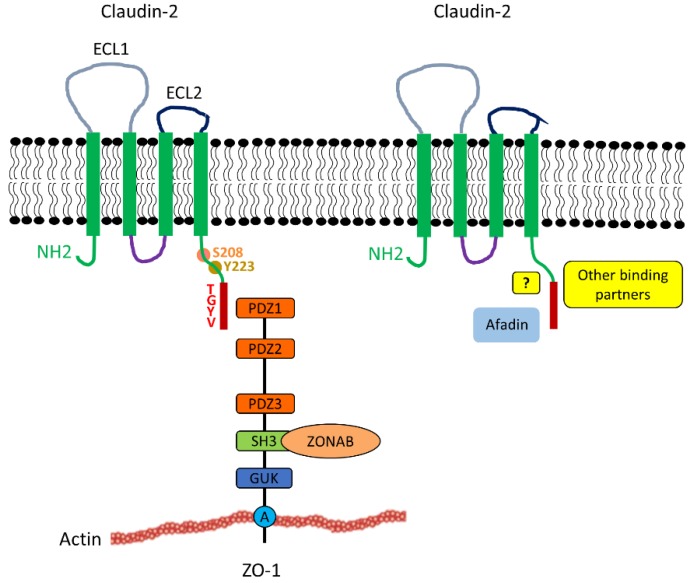
Claudin-2 structure and interactions with cytosolic multidomain adapters. Claudin-2 consists of two extracellular loops, the longer ECL1 (grey) and shorter ECL2 (black). Claudin-2 also contains 4 transmembrane domains (green box), a cytoplasmic loop (purple), a short N-terminal region and a longer C-terminal region (green). Claudin-2 interacts with the ZO family of TJ plaque proteins (for clarity only ZO1 is depicted) through its PDZ-binding motif - TGYV located at the end of the C-terminus (indicated by the red box). The domains of ZO1 depicted include PDZ1, PDZ2, and SH3, mediating binding of a variety of proteins to create large multiprotein complexes; GUK and A, for actin binding segment. The SH3 domain of ZO-1 was shown to bind to the transcription factor ZONAB, which may play a role in the proliferative effects of claudin-2 (Section 4.2). Afadin is a recently identified claudin-2 partner. The mode of coupling (direct biding or indirect association through adapters) remains to be established. Other newly identified candidate binding partners for claudin-2 include Scrib, Arhgap21, PDLIM2/7, and Rims-2 [34]. The claudin-2 tail contains many potential phosphorylation sites. Among these, Y223 affects the affinity of the PDZ binding domain [35], and S208 appears to be a switch for membrane retention and lysosomal or nuclear localization (see Section 3.4 and Section 4.2).

**Figure 2 ijms-20-05655-f002:**
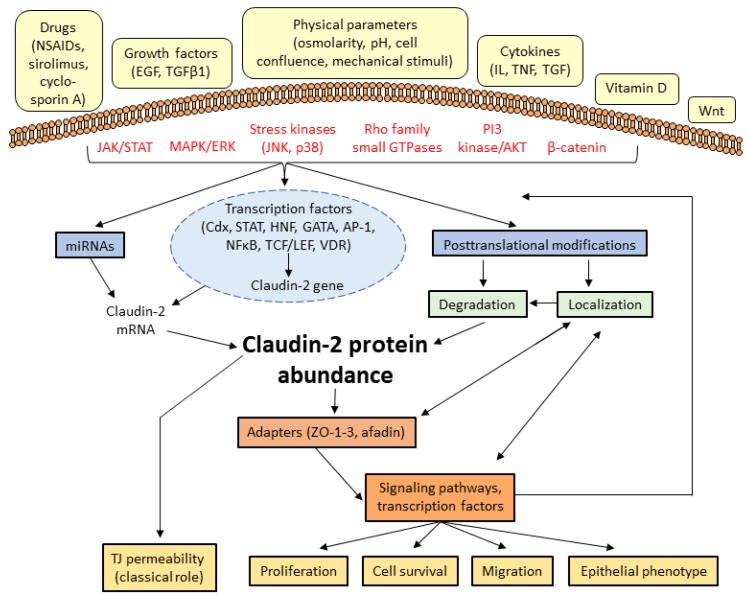
Schematic overview of claudin-2 regulation and its downstream effects. Various extracellular stimuli activate signaling pathways that impact claudin-2 expression by altering its synthesis, and via post-transcriptional and post-translational modifications. In addition to its permeability effects, altered claudin-2 expression also modulates various cellular processes likely acting via a number of downstream signaling pathways and transcription factors (see Table 1 for details on the signaling and effects downstream from claudin-2).

**Table 1 ijms-20-05655-t001:** Summary of the experimental evidence for a causal link between claudin-2 expression and functional outcomes.

Cell Line or Transgenic Mouse	Change in Claudin-2 Expression	Downstream Functional Effect	Signaling Components Downstream from Claudin-2	Ref.
Villin-claudin-2 transgenic mice	Claudin-2 overexpression	Increased colonocyte proliferation	PI-3K/Bcl-2 pathway	[116]
Caco2 human colon cancer cell	Claudin-2 silencing	Reduced EGF-induced proliferation		[51]
SW480 and HCT116 colon cancer cell lines	Claudin-2 overexpression	Increased proliferation and anchorage-independent growth		
LLC-PK1 porcine kidney tubular cells	Claudin-2 silencing	Reduced proliferation	GEF-H1-mediated RhoA activation, increase in p27^kip1^	[67]
		Pro-fibrotic epithelial shift	RhoA-mediated MRTF activation	
A549 human lung adenocarcinoma cells	Claudin-2 downregulation/ knockdown	Reduced G1/S transition	Cyclin D1 and E1, ZONAB	[103]
		Increased sensitivity to anti-cancer agents; increased intracellular drug accumulation and reduced efflux	Decrease in phosho-c-Jun and nuclear Sp1; reduced expression of multidrug resistance-associated protein/ABCC2	[119]
		Reduced migration	Decreased Sp1, reduced MMP-9 expression, activity	[120]
A549 human lung adenocarcinoma cells	Flavonoid- or epigenetic inhibitor-induced claudin-2 downregulation;	Reduced proliferation; partial rescue of phenotype by claudin-2 reexpression		[64,117]
HT-29 colorectal cancer cell line	Symplekin silencing-induced claudin-2 downregulation;	Reduced anchorage-dependent growth and Zonab nuclear localization; rescue of phenotype by claudin-2 reexpression		[118]
A549 human lung adenocarcinoma cells	Endocytosis and lysosomal degradation of claudin-2 induced by a peptide mimic (DFYSP) of a conserved ECL2 region	Claudin-2 accumulation in the lysosomes, cellular injury and necrotic cell death	Cathepsin B release from lysosomes	[40]
MDCK canine tubular cells	Inducible knockdown of claudin-2	Enhanced migration in a wound-healing assay	Increased MMP-9 mRNA and activity	[120]
Human colon cancer stem-like cells (patient-derived CCP1 cells)	Claudin-2 overexpression	Self-renewal of cancer stem cells; enhanced tumor initiation, progression, and metastasis	YAP and miRNAs (especially miR-222-3p)	[122]
Caco2 human colon cancer cell	Claudin-2 overexpression	Enhanced migration	Effect independent from MMP-2 and 9	[128]
AGS stomach carcinoma cells	Claudin-2 overexpression	Enhanced migration	Likely via increased MMP-1, -2 and 9 expression	[125]
T-84 colonic adenocarcinoma, AGS and KATO-III stomach carcinoma cells; A549 lung adenocarcinoma cell lines	Claudin-2 silencing	Reduced migration		[125]
	Claudin-2 overexpression	Augmented migration		
	Non-steroidal anti-inflammatory drugs (NSAIDs)-induced claudin-2 downregulation	Reduced migration; claudin-2 reexpression counteracted the effect		
U2OS osteosarcoma cell line	Claudin-2 overexpression	Reduced migration and invasion	Afadin-mediated ERK inhibition	[130]
Fetal osteoblast cell line hFOB.1.19	Claudin-2 silencing	Augmented migration and invasion	ERK activation, afadin reduction	
Claudin-2 KO mice	Claudin-2 KO	Augmented TNFα-induced colorectal inflammation	NFκB, myosin light chain kinase	[135]
Claudin-2 KO mice	Claudin-2 KO	Augmented energy demand of transport processes; increased ischemia-reperfusion kidney injury		[110]

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
