# Peer review of "Claudin-2: Roles beyond Permeability Functions"

_ijms, 2019, doi:10.3390/ijms20225655_

Round 1

Reviewer 1 Report

The review entitled “Claudin-2: roles beyond permeability functions” by Venugopal, Anwer and Szászi is a very thorough review of claudin-2 function and regulation.  The authors should be congratulated on a well-written and complete review of the claudin-2 literature. 

Minor suggestions to improve the manuscript are listed below.

Final review for typographical/editing changes to address some missing periods, closing parentheses, format of subheadings, etc.Also, throughout make sure when referring to “next section” or later in review to give precise section number. This is done in some places but not all. Throughout the manuscript, the authors never refer to which species they are studying, we assume human for most instances.That being said it would be helpful to indicate this when saying that there are 27 claudin family members (line 41) as the number varies between species. Also instead of the 2012 review (ref 4), it might be more appropriate to cite  a more recent review such as: Tsukita, S; Tanaka, H; Tamura, A. 2019. The Claudins: From tight junctions to biological systems. Trends Biochem Sci, 44, 141-152. Somewhere in the review it would be appropriate to include how conserved claudin-2 is between species. Line 74-115: In section 2.2 there is no reference towards the importance, relevance or function of ECL2. This domain is usually involved in trans-interactions and should not be left out. As indicated by the authors reference 27 show that claudin-2 homodimerizes by ECL1; however, what about trans-interactions with other claudin members?How does it interact with Cldn3? If ECL2 is not involved in interaction then what it is its function?   Lines 76/77 – contains the comment that ‘some studies found the specific claudin-2 band at a lower molecular weight’. If not common in all then perhaps there should be mention of what other studies see or if this is dependent on the antibody reagent or cell-type being examined.This would be valuable information for the reader, particularly those who might be new to studying claudin-2.  Lines 112-115: Consider balance of the information provided here.Several post-translational modifications are mentioned but then a specific example is given for phosphorylation, while further details from the other references only come up later.  Either all can be later or all discussed here.  On Line 114 the authors refer to Tyrosine at position -6.This is a bit confusing for several reasons.  Many will read Tyr-6 and assume it is the 6th amino acid from the N-terminus.  Also, although the manuscript by Nomme et al., 2015 do refer to this Tyrosine as being at -6 from the C-terminus it is actually the 7th amino acid from the C-terminus.  This becomes even more confusing in Especially in lines 260-265 where S208 and Y-6 are both discussed.  Since this review is discussing only claudin-2 it would be much clearer to give the actual position of this amino acid and then say what species you are referring to if it changes between species. Line 165 – Add ‘context-dependent’ to describe discrepancies.The rest of this paragraph makes it clear, however, it should be clear from the start that these discrepancies are context-dependent as compared to discrepancies in the same conditions/cell lines between data generated from different groups. Review section on the section on signal transduction and claudin-2 expression. Is this section mostly about protein levels and localization or also transcriptional effects? Not very clear. Line 260-270: The discussion regarding post-translational modifications is interesting. The authors are encouraged to ensure that the information provided is complete.For example, reference 98 explores the role of phosphorylation of claudin-1 and -2 and membrane availability/degradation. Also, how does ref 98 reconcile with ref 36? Ref 36 shows phosphorylation promotes membrane retention and reduced lysosome trafficking of claudin-2, while ref 98 shows that phosphorylation increases claudin-2 degradation.  In this scenario, as the author suggest on line 359, does de-phosphorylation of  claudin-2 promote nuclear translocation (ref 107) or does it stabilize claudin-2 at the membrane (ref 98)? In the section that discusses the role of claudin-2 in cancer, it would be interesting to have a short subsection about the role of claudin-2 in metastasis given the increasing data about claudins in metastasis and migrating cancer cells.Some of these points are already included in the review but it might could to consolidate this information. In Figure 1 legend – describe use of colour to indicated different claudin domains. As shown, it looks like S208 is potentially involved in Afadin binding but not in other interactions.The figure legend describes the function of S208 but these functions are not obvious in the figure. Relevance of showing all three ZO proteins is not clear given that interaction is the same. Yet as diagrammed it looks like PDZ domain is important for interaction with ZO-1 but COOH domain in general is important for  interaction with ZO-3. It is appreciated that including all of the concepts shown in Figure 2 is challenging.However, this figure would be more useful if direction (up or down) of regulation, modification, localization, etc. was clear. As is, it gives the impression that we do not understand the effects of any of the signaling pathways – all seem to affect Claudin-2 expression at the level of the gene, miRNA, protein stability/localization – or the downstream pathways. It is suggested that the authors consider if this information would be more meaningful if presented in a modified figure of a table.  For example, columns in a table might include “category of signal”, “effectors”, “level of effect (and direction) on claudin-2 expression”, “downstream effect”

Author Response

Response to reviewer 1.

We thank the reviewer for the thorough report on our paper and the excellent suggestions. We made a thorough revision to address all points, as summarized below.

1, Final review for typographical/editing changes to address some missing periods, closing parentheses, format of subheadings, etc.

Answer: We did a thorough proof reading. The template used was originally designed for experimental papers, and therefore we adapted it to our needs, while ensuring that the subheadings were consistent throughout the text.

2, Also, throughout make sure when referring to “next section” or later in review to give precise section number. This is done in some places but not all.

Answer: We provided section numbers in all of our cross references

3, Throughout the manuscript, the authors never refer to which species they are studying, we assume human for most instances. That being said it would be helpful to indicate this when saying that there are 27 claudin family members (line 41) as the number varies between species.

Answer: We added a brief description on the species differences of claudins and indicated that the review focuses on mammalian claudins. We also noted homology in claudin-2 between mouse and human (Line 40-46). Further, in the places where it is highly relevant, we also mention the species (e.g. human) of the protein discussed.

4, Also instead of the 2012 review (ref 4), it might be more appropriate to cite  a more recent review such as: Tsukita, S; Tanaka, H; Tamura, A. 2019. The Claudins: From tight junctions to biological systems. Trends Biochem Sci, 44, 141-152.

Answer: Thank you for this suggestion, we added this excellent recent review.

5, Somewhere in the review it would be appropriate to include how conserved claudin-2 is between species.

Answer: We added information on this (see above).

6, Line 74-115: In section 2.2 there is no reference towards the importance, relevance or function of ECL2. This domain is usually involved in trans-interactions and should not be left out. As indicated by the authors reference 27 show that claudin-2 homodimerizes by ECL1; however, what about trans-interactions with other claudin members? How does it interact with Cldn3? If ECL2 is not involved in interaction then what it is its function?    

Answer: There is limited information available on the function of ECL2. It appears to be different from some other claudins (e.g. claudin-5) in that it does not mediate homotypic interactions. The exact structural requirements for the trans interactions of claudin-2 are not clear . This is not stated more explicitly in the indicated section (lines 102-108)

7, Lines 76/77 – contains the comment that ‘some studies found the specific claudin-2 band at a lower molecular weight’. If not common in all then perhaps there should be mention of what other studies see or if this is dependent on the antibody reagent or cell-type being examined. This would be valuable information for the reader, particularly those who might be new to studying claudin-2. 

Answer: We agree that this would have been valuable information, however when we started collecting this info, we realized that in many publications the molecular weight is not indicated and/or the exact antibody used is not clear. While this info would be useful, we could not assemble the information in the available time frame. Therefor we removed this comment.

Lines 112-115: Consider balance of the information provided here. Several post-translational modifications are mentioned but then a specific example is given for phosphorylation, while further details from the other references only come up later.  Either all can be later or all discussed here. 

Answer: We deleted the sentence the reviewer referred to, and merged the information into the section discussing the effect of phosphorylation on trafficking (starting at line line 247).

9, On Line 114 the authors refer to Tyrosine at position -6. This is a bit confusing for several reasons.  Many will read Tyr-6 and assume it is the 6th amino acid from the N-terminus.  Also, although the manuscript by Nomme et al., 2015 do refer to this Tyrosine as being at -6 from the C-terminus it is actually the 7th amino acid from the C-terminus.  This becomes even more confusing in Especially in lines 260-265 where S208 and Y-6 are both discussed.  Since this review is discussing only claudin-2 it would be much clearer to give the actual position of this amino acid and then say what species you are referring to if it changes between species.  

Answer: As mentioned above, we merged the two sections. We also clarified the numbering and now refer to this tyrosine as Y224. We also clarify that these data refer to the human claudin (line 246 onward).

10, Line 165 – Add ‘context-dependent’ to describe discrepancies. The rest of this paragraph makes it clear, however, it should be clear from the start that these discrepancies are context-dependent as compared to discrepancies in the same conditions/cell lines between data generated from different groups.

Answer: This was added (now line 140).

11, Review section on the section on signal transduction and claudin-2 expression. Is this section mostly about protein levels and localization or also transcriptional effects? Not very clear.

Answer: A brief section was added to explain the intertwined nature of the effects of these signalling pathways, and to clarify the topic (transcriptional effects vs other effects) (line 159-163)

12, Line 260-270: The discussion regarding post-translational modifications is interesting. The authors are encouraged to ensure that the information provided is complete. For example, reference 98 explores the role of phosphorylation of claudin-1 and -2 and membrane availability/degradation. Also, how does ref 98 reconcile with ref 36? Ref 36 shows phosphorylation promotes membrane retention and reduced lysosome trafficking of claudin-2, while ref 98 shows that phosphorylation increases claudin-2 degradation.  In this scenario, as the author suggest on line 359, does de-phosphorylation of claudin-2 promote nuclear translocation (ref 107) or does it stabilize claudin-2 at the membrane (ref 98)?

Answer: We consolidated the discussion on the role of S208 into the section stating at line 252 (section 3.4). We extended the discussion that now includes ref 98 (now ref 47). In fact, the papers discussed appear to be in good agreement, showing that under various conditions on tubular cells, the non-phosphorylatable mutant S208A is more cytosolic and lysosomal, while the phospho-mimetic mutant S208E found more at the TJs.  In addition, the S208A mutant was also found to be more nuclear (ref 102) lending credence to the notion that dephosphorylation of claudin-2 might induce its removal from the TJs and promote both lysosomal and nuclear localization.    

13, In the section that discusses the role of claudin-2 in cancer, it would be interesting to have a short subsection about the role of claudin-2 in metastasis given the increasing data about claudins in metastasis and migrating cancer cells. Some of these points are already included in the review but it might could to consolidate this information.

Answer: As suggested, we created a separate section discussing the role of claudin-2 in metastasis (line 566). We also changed the relevant subsection title to Claudin-2 in cancer and metastasis formation.

14, In Figure 1 legend – describe use of colour to indicated different claudin domains.

Answer: We added this info to the legend.

15, As shown, it looks like S208 is potentially involved in Afadin binding but not in other interactions.

Answer: We changed the figure to clarify this.

16, The figure legend describes the function of S208 but these functions are not obvious in the figure. Relevance of showing all three ZO proteins is not clear given that interaction is the same.

Answer: We added a line to the legend to briefly describe finction and refer to the corresponding section.

17, Yet as diagrammed it looks like PDZ domain is important for interaction with ZO-1 but COOH domain in general is important for interaction with ZO-3.

Answer: We simplified the figure and now only show ZO1. We explain in the legend that  for the sake of clarity on ZO1 is shown.

18, It is appreciated that including all of the concepts shown in Figure 2 is challenging. However, this figure would be more useful if direction (up or down) of regulation, modification, localization, etc. was clear. As is, it gives the impression that we do not understand the effects of any of the signaling pathways – all seem to affect Claudin-2 expression at the level of the gene, miRNA, protein stability/localization – or the downstream pathways. It is suggested that the authors consider if this information would be more meaningful if presented in a modified figure of a table.  For example, columns in a table might include “category of signal”, “effectors”, “level of effect (and direction) on claudin-2 expression”, “downstream effect”

Answer: Thank you for this excellent suggestion. We generated a table to summarize the experimental evidence available on the signal modulating effects of claudin-2 (Table 1). We included only data where experimental verification for the causal relationship was available (i.e. where functions downstream from direct modulation of  claudin-2 expression were studies; or where retransfection of the protein verified its causal role), and excluded the correlative data, which would have made the table too large. The table lists the signalling components that were suggested to connect claudin-2 with downstream effects (but for clarity, we did not include signals upstream from claudin-2). We kept the original figure, but included in the legend a line referring to the table. Since the data on claudin-2 regulation is substantial, we could not generate a similar table due to time constraints for the revision. We also felt that such a table would be too large.

Reviewer 2 Report

This is a well-written and comprehensive review that focuses on an important epithelial protein, caludin-2. Unlike previous paper that summarise functions of this tight junction protein, the current review sought to describe the roles of claudin-2 that go beyond regulation of paracellular permeability. I believe the authors successfully achieved this goal.

Generally, the paper present an important, up to date information which is well-described and discussed. However, on my opinion, organization of the manuscript is not optimal and could be improved in the following way:

I believe all functions of claudin-2 should be discussed together. Therefore I would recommend to move the subchapter 2.3 that discussed claudin-dependent paracellular permeability and place it after Chapter 3 that describes regulation of claudin-2 expression.  The subchapter 4.1 Emerging mechanisms is also misplaced and should go after describing all cladin-2 functions (regulation of permeability, proliferation and cell motility). In fact, this chapter outlines molecular mechanisms that may regulate aforementioned claudin functions. Lines 282-297  describe a single paper about novel interacting partners of claudin-2 that have been already described in a different part of the manuscript (Subchpter 2.2). Line 73. What is epiphenomenon? 

Author Response

Response to Reviewer 2

We thank the reviewer for the praise and for the excellent suggestions. The revision addresses all points raised, as summarized below.

1, I believe all functions of claudin-2 should be discussed together. Therefore I would recommend to move the subchapter 2.3 that discussed claudin-dependent paracellular permeability and place it after Chapter 3 that describes regulation of claudin-2 expression.  The subchapter 4.1 Emerging mechanisms is also misplaced and should go after describing all cladin-2 functions (regulation of permeability, proliferation and cell motility). In fact, this chapter outlines molecular mechanisms that may regulate aforementioned claudin functions.

Answer: We followed the advice of the reviewer and changed the order of description, as recommended.

2, Lines 282-297  describe a single paper about novel interacting partners of claudin-2 that have been already described in a different part of the manuscript (Subchpter 2.2). Line 73.

Answer: We removed the details from section 2.2. and merged all information into one section.

3, What is epiphenomenon? 

Answer: We removed the expression from one of the original sentences where it was used. In the section starting at line 481 we refined the explanation in the sentence to better explain what is meant by epiphenomenon; “Initially, the described claudin expression alterations were considered an epiphenomenon; i.e. they were assumed to arise due to the underlying disease process, but were thought not to be causally contributing to pathogenesis”